# Genome skimming reveals novel plastid markers for the molecular identification of illegally logged African timber species

**Maurizio Mascarello**[1,2]*, **Mario Amalfi**[1,3], **Pieter Asselman**[4], **Erik Smets**[5], **Olivier J. Hardy**[6], **Hans Beeckman**[7], **Steven B. Janssens**[1,2]

**1** Meise Botanic Garden, Meise, Belgium, **2** Department of Biology, KU Leuven, Leuven, Belgium, **3** Fédération Wallonie–Bruxelles, Service général de l'Enseignement universitaire et de la Recherche scientifique, Brussels, Belgium, **4** Mycology & Systematic and Evolutionary Botany, Department of Biology, Ghent University, Ghent, Belgium, **5** Naturalis Biodiversity Center, Leiden, The Netherlands, **6** Evolutionary Biology and Ecology, Université Libre de Bruxelles, Brussels, Belgium, **7** Wood Biology, Department of Biology, Royal Museum for Central Africa, Tervuren, Belgium

* maurizio.mascarello.88@gmail.com

## Abstract

Tropical forests represent vast carbon stocks and continue to be key carbon sinks and buffer climate changes. The international policy constructed several mechanisms aiming at conservation and sustainable use of these forests. Illegal logging is an important threat of forests, especially in the tropics. Several laws and regulations have been set up to combat illegal timber trade. Despite significant enforcement efforts of these regulations, illegal logging continues to be a serious problem and impacts for the functioning of the forest ecosystem and global biodiversity in the tropics. Microscopic analysis of wood samples and the use of conventional plant DNA barcodes often do not allow to distinguish closely-related species. The use of novel molecular technologies could make an important contribution for the identification of tree species. In this study, we used high-throughput sequencing technologies and bioinformatics tools to obtain the complete de-novo chloroplast genome of 62 commercial African timber species using the genome skimming method. Then, we performed a comparative genomic analysis that revealed new candidate genetic regions for the discrimination of closely-related species. We concluded that genome skimming is a promising method for the development of plant genetic markers to combat illegal logging activities supporting CITES, FLEGT and the EU Timber Regulation.

## Introduction

Strong regulations and international treaties were established to control forest logging in tropical regions [1]. However, it is estimated that up to 90% of the annual timber volume is obtained through illegal logging activities in tropical countries such as the Democratic Republic of the Congo [2, 3]. Although the Convention on International Trade in Endangered Species of Wild Fauna and Flora (CITES; www.cites.org) lists approximately 350 tree species

**Data Availability Statement:** All relevant data are within the paper and its Supporting Information

files. All relevant Genbank accession numbers are also within the Supporting Information files.

**Funding:** This study is supported by the Plant.ID project. Plant.ID has received funding from the European Union's Horizon 2020 research and innovation programme under the Marie Skłodowska-Curie grant agreement No 765000. In addition, this research received support from the SYNTHESYS Plus project (https://www.synthesys.info/) funded under H2020-EU.1.4.1.2. grant agreement 823827.

**Competing interests:** The authors have declared that no competing interests exist.

worldwide, CITES regulations can only be effective when a synergy between international law enforcement and an efficient police and investigative force in the importing countries is established. In case this system is not sufficiently effective, corruption and laundering of illegally harvested wood can easily take place.

In addition to CITES, further initiatives were launched in the 2000's to combat illegal logging and trade in wood products [3]. Firstly, the European Union Forest Law Enforcement, Governance and Trade (EU FLEGT) Action Plan (http://www.euflegt.efi.int) was established in 2003 as bilateral negotiation between the European Union (EU) and timber-exporting countries to prohibits the trade and import of illegal timber by the EU. Further legislations to cope with illegal timber trade include the EU Timber Regulations (EUTR) and the Voluntary Partnership Agreements (VPAs), entered into force in 2003 and 2011, respectively. Central African countries such as Cameroon, Central African Republic and the Republic of Congo already signed a VPA with the European Union. A further initiative is an extension of the United States Lacey Act (https://www.fws.gov) in 2008, which illegitimates wood harvesting, processing and trade when occurring in violation of the United States law.

Log tracking is the main strategy applied to avoid laundering of illegal timber and refraining from false certificates [1]. This cost-effective method allows to track timber from the time of harvesting to its arrival to the processing plant, and it is based on the detection of wood through conventional and ultraviolet tracking paints. However, log tracking is often not flawless because of corruption and the application of several strategies aimed to overcome controls by both the local police and international authorities [4]. These include logging beyond concessions, falsification of export permits and mixing illegally logged timber with wood obtained via official procedures [5–7]. As a result, there is a substantial need for more advanced forensic techniques to support the identification of logs and wood products.

The first method used in forensic laboratories is the microscopical identification of wood anatomical characters [8], which often leads to the successful discrimination between different genera, and in some cases also between species [9]. However, forensic wood anatomical techniques are often limited in distinguishing closely-related species when they are characterized by similar wood properties. As a consequence, closely related species that differ in conservation status are sometimes indifferently mixed in timber trade [8]. A more recent tool concerns the application of molecular markers as a promising method for reliable and cost-effective identification of plant species. This technique is known as DNA barcoding, which applies the use of short DNA sequences for species identification; i.e. 'barcodes'. Conventional plant barcodes include the coding regions of the plastid genes *matK* and *rbcL*, the intergenic spacer *trnH-psbA*, the *trnL* intron, as well as the nuclear ribosomal internal transcribed spacers (ITS1 and ITS2) [10–14]. However, it recently became clear that these standard plastid markers often fail to provide successful discrimination power among closely-related species [14, 15]. Although ITS1 & ITS2 provide a higher species identification rate, this often results in a scarce resolution due to the presence of multiple copies in cells of the nuclear ribosomal RNA. Moreover, primers used to amplify plant and fungal nrITSs are similar, leading to sample misidentification due to fungal contaminations [16]. The limitation of standard forensic techniques requires novel, efficient and cost-effective molecular techniques which could improve the identification of a large number of commercial timber species. These would be useful to support the enforcement of the forestry law and of conservation systems for the conservation of the tropical forest, detecting the illegal trade of protected timber species [17, 18].

In this study, we want to evaluate the potential of a novel molecular high throughput sequencing technique, namely "genome skimming", on a set of commercial timber species that occur in Central and West tropical Africa. This method consists in sequencing short DNA fragments at low sequence coverage (0.1X-10X) in order to obtain a large number of raw reads

for the *de-novo* reconstruction of high-copy organelle genomes, including the plastid and mitochondrial genomes, as well as high-copy nuclear ribosomal DNA [17, 19, 20]. Genome skimming increases the chance to recover entire genetic sequences in a fast and cost-effective manner, which could not be amplified via traditional Sanger sequencing. As a result, this technique is also suitable for the retrieval of high-copy organelle sequences and genome reconstruction from degraded DNA material because 100–200 bp long DNA fragments are sufficient [21]. We focus on the chloroplast genome because it exhibits a conserved gene order and a low substitution rate [19]. In addition, the plastome was found to be suitable for the differentiation of closely-related species. For example, Bi et al. [22] and Song et al. [23] identified putative molecular markers in the chloroplast genomes within a group of *Fritillaria* and *Dalbergia* species, respectively, which could be useful for the development of DNA barcodes for species discrimination. Furthermore, Malé et al. [24] used the chloroplast genome to resolve the phylogeny of a group of eight species belonging to the Chrysobalanaceae family, obtaining a major resolution in comparison with that obtained upon the analysis of the mitochondrial and nuclear ribosomal genomes.

The aim of this study is to process DNA samples of major commercial African timber species obtained from dried plant material, and to sequence them through High-Throughput Sequencing (HTS) technologies (Illumina sequencing), in order to reconstruct the complete chloroplast genome using bioinformatics tools. In this way, we search for candidate genetic markers by the identification of highly variable polymorphic sites in the chloroplast genome.

## Material and methods

### Plant sampling

This study includes 62 African timber species belonging to six different families in the angiosperms (S1 Table). These include the main commercial timber species in Central and West tropical Africa which are exploited for their highly valuable timber, and which are additionally reported in treaties/agreements for the conservation of the global biodiversity, such as the CITES and the International Union for Conservation of Nature (IUCN; https://www.iucnredlist.org/). Besides these species, we also included tree species which could gain economic importance in the timber trade and those for which logging operations could constitute a future threat. Finally, we also focused on species that are closely related to the first two categories and could be included in mixed consignment with the timber species of major interest in the timber market. Leaf samples from each species were collected from herbarium specimens and silica-dried material obtained from different herbaria (S1 Table). The sample Van der Burgt 1932 (*Lophira alata*) was collected after the emendation of the Nagoya Protocol for Access and Benefit-Sharing (www.cbd.int), entered into force on 12 October 2014. We received all permits (collection: OFC 769 (H2016/00223)) from the Royal Botanic Garden, Kew (UK).

### DNA extraction and library preparation

DNA isolation was performed using a CTAB extraction protocol in combination with additional sorbitol washing, chloroform/isoamyl alcohol 24:1 extraction and isopropanol precipitation [25]. DNA samples were finally eluted using 1X TE buffer. DNA purity was assessed by measuring the absorbance ratio (OD) 260/280 and the OD 260/230 values using NanoDrop™ 2000 (Thermo Fisher Scientific, US). DNA concentration (ng/μl) and fragment size distribution were measured by capillary electrophoresis using Fragment Analyzer (Agilent, US).

DNA libraries were prepared through a starting enzymatic DNA fragmentation (to obtain a fragment size distribution of 200–450 bp), and end repair using NEBnext® Ultra™ II FS DNA

Library Prep Kit for Illumina® (New England Biolabs, US). Highly-degraded DNA samples exhibiting low values of average fragment size (bp) were end-repaired without prior DNA fragmentation using NEBnext® Ultra™ II DNA Library Prep Kit for Illumina® (New England Biolabs, US). Then, additional adapter ligation and U-excision were performed using NEBnext® Adaptor for Illumina® and USER® Enzyme (New England Biolabs, US), respectively. DNA library size selection (320–470 bp) was performed using SPRIselect® (Beckman Coulter, US). Adaptor-ligated DNA was indexed and PCR-enriched using NEBNext® Ultra II Q5 Master Mix and NEBNext® Multiplex Oligos for Illumina® (New England Biolabs, US). The thermocycler was set-up as follow: 1) Initial denaturation (98˚C, 30 s), 2) 3–4 cycles of denaturation (98˚C, 10 s) and annealing/extension (65˚C, 75 s), 3) Final extension (65˚C, 5 min). A final DNA-library purification was performed using SPRIselect® (Beckman Coulter, US). Degraded DNA samples were processed without initial enzymatic DNA fragmentation using NEBnext® Ultra™ II DNA Library Prep Kit for Illumina® (New England Biolabs, US). The final fragment size distribution and molarity (nM) were checked by loading the DNA samples on a Fragment Analyzer (Agilent, US). Finally, DNA libraries were sent to RAPiD Genomics (Florida, US) and Fasteris SA (Plan-les-Ouates, Switzerland) for low coverage paired-end sequencing (10X, 150 bp) sequencing, performed using HiSeq® 3000, as well as HiSeq® 4000 and NovaSeq® 6000 (Illumina, US).

**Genome assembly and annotation.** Raw reads quality was checked using FastQC [26]. Chloroplast genomes were *de-novo* assembled using the automatic GetOrganelle pipeline [27], which recruits plastid-like reads using Bowtie2 [28], trim reads and reconstructs contigs using SPAdes 3.13 [29], and filters plastid-like contig against the BLAST nucleotide database using NCBI Blast+ [30]. The reconstructed genomes were firstly aligned with a closed reference genome from GenBank (http://www.ncbi.nlm.nih.gov/genbank/) using MAFFT v.7 [31]. The orientation of flip-flop sequences was further re-arranged using Geneious Prime [32]. The genetic regions which were not successfully assembled were retrieved by mapping the raw reads to the target regions of closed-related species using Bowtie 2 (local alignment). Genome annotation was carried out using the web-based software CpGAVAS2 [33]. The resulting annotations were validated in Geneious Prime by comparison with the closed reference plastid genome downloaded from GenBank (S2 Table).

## Identification of highly variable regions for species discrimination

Chloroplast genomes were aligned at the family level using MAFFT v.7. Then, we visually inspected nucleotide variations between species and calculated their pairwise similarity in Geneious Prime. Species exhibiting a difference of a maximum of both 5 *single-nucleotide polymorphisms* (SNPs) in the *matK* coding region and 3 SNPs in the *rbcL* coding region were re-aligned separately. Then, the resulting multiple alignments were manually adjusted in Geneious Prime. After then, we reversed sequence inversions in each multiple alignment in order to highlight regions presenting a high number of SNPs. Finally, the nucleotide variability (Pi) and the number of polymorphic sites (S) were calculated in DnaSP v.6.12.03 [34] using a sliding window analysis (window length = 600 bp; step size = 200 bp).

## Results

### Chloroplast structure

Using the Illumina approach following a 'genome skimming' methodology, genomic reads of high-copy organelle genomes (mt, cp and nr) were retrieved for 62 African tree species. Based on those raw sequence reads, the chloroplast genome of each species was *de novo* reconstructed (S1 File) and compared to reference plastomes found in GenBank (S2 Table).

Plastomes of all species analyzed in this study consist of a circular sequence that is partitioned in four main structures typical for land plant chloroplasts: the large (LSC) and small (SSC) single copy regions, separated by two inverted repeats (IRa and IRb) (Fig 1). At the family level, the subfamily Caesalpinioideae exhibits the highest plastome length (160674 bp ± 648 bp), while Meliaceae presents the highest GC content (37,8% ± 0.1%) (Table 1). At genus level, *Millettia* (Fabaceae) exhibits both the lowest plastome length (154245 bp ± 61 bp) and the lowest GC values (35.2% ± 0.1%), while *Lovoa* (Meliaceae) is characterized by the highest GC values (38%) (S3 Table).

## Gene order and content

Plastomes obtained in this study consist of up to 113 unique genes, of which 79 are coding sequences, 30 are tRNAs and 4 are rRNAs (Table 2). Up to 18 genes from the IRb region are duplicates of those that occur in IRa. However, only 17 duplicate genes are located in the IRa of species belonging to Fabaceae-Papilionoideae, Ebenaceae, Sapotaceae, and in *Morus mesozygia*, as the *rps19* is absent in this region. Furthermore, the *rps19* occurs in the IRa as pseudogene in Fabaceae-Detarioideae, Fabaceae-Caesalpinioideae, and in *Milicia* species, while it exhibits a complete coding region in all Meliaceae species. 18 genes contain intron sequences, including six tRNA genes and 12 coding regions. The *clpP*, *ycf3* and *rps12* genes have two intron regions. In the *rps12* gene, the 3'*rps12* (exon 2 and exon 3) are trans-spliced to the 5'*rps12* (exon 1) [36, 37]. In addition, as previously reported by Gantt et al. [38], the coding gene *rpl22* is not expressed in Fabaceae species. Furthermore, the coding gene *infA* lost its function in all species belonging to the rosids (Fabaceae, Meliaceae, Moraceae and Ochnaceae). However, *infA* is still expressed in species of the Asterid clade (Ebenaceae and Sapotaceae).

The gene order is conserved and identical for the majority of the species. However, we confirm that all species belonging to the subfamily Papilionoideae (Fabaceae family) exhibit a 50-kb inversion in the LSC region situated between the *matK* and *accD* genes [40, 41]. In addition, we observed that, similar to *Millettia pinnata* [42], *Millettia* species are also characterized

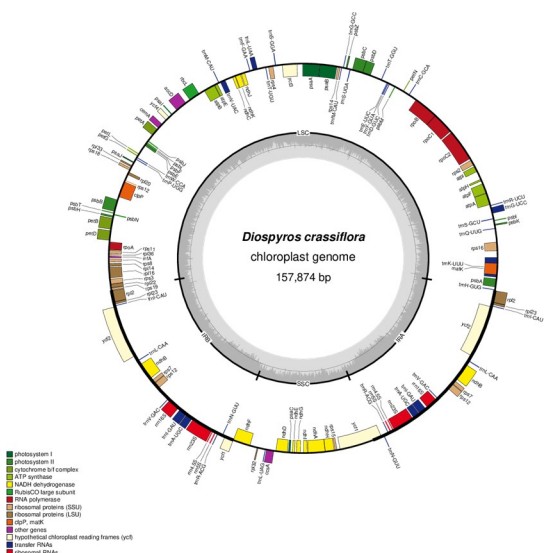

**Fig 1. Complete chloroplast genome of *Diospyros crassiflora* including gene annotation and GC content.** The gene map was created using OrganellarGenomeDRAW [35].

**Table 1. Average plastome length (bp) and GC content (%) values at the family and subfamily levels (of the species studied here), and their respective standard deviations.**

| Family | Plastome length (bp) | GC content (%) |
|---|---|---|
| Ebenaceae (N = 2) | 157560 ± 444 | 37.4 ± 0.0 |
| Fabaceae-Caesalpinioideae (N = 5) | 160674 ± 648 | 36.4 ± 0.1 |
| Fabaceae-Detarioideae (N = 14) | 159792 ± 637 | 36.1 ± 0.2 |
| Fabacceae-Papilionoideae (N = 10) | 157742 ± 2284 | 36 ± 0.5 |
| Meliaceae (N = 14) | 159822 ± 456 | 37.8 ± 0.1 |
| Moraceae (N = 3) | 160127 ± 109 | 36 ± 0.0 |
| Ochnaceae (N = 3) | 159113 ± 1359 | 36.4 ± 0.1 |
| Sapotaceae-Chrysophylloideae (N = 7) | 158975 ± 133 | 36.7 ± 0.1 |
| Sapotaceae-Sapotoideae (N = 4) | 159649 ± 257 | 36.8 ± 0.0 |

by a 6.5kb inversion of the region between *trnG-GCC* and *ycf4*, located in the second junction of the 50kb inversion of Papilionoideae. Finally, *Pericopsis* species contain a 2-kb inversion of the region located between *psbE* and the *psaJ*. As a result, the *trnP* and *trnW* genes are oriented in the opposite direction, while the *petG* and *petL* genes are transcribed in the reverse direction. Interestingly, *Bobgunnia* species that also occur in the Papilionoideae group, where they are part of the Swartzieae tribe [41], do not show the 50kb inversion of the LSC region.

**Table 2. List of chloroplast genes and their respective biological function.**

| Function | Gene |
|---|---|
| RNAs, ribosomal | *rrn23, rrn16, rrn5, rrn4.5* |
| RNAs, transfer | *trnA-UGC*\*, *trnC-GCA, trnD-GUC, trnE-UUC, trnF-GAA, trnG-GCC, trnG-UCC*\*, *trnH-GUG, trnI-CAU, trnI-GAU*\*, *trnK-UUU*\*, *trnL-CAA, trnL-UAA*\*, *trnL-UAG, trnM-CAU, trnfM-CAU, trnN-GUU, trnP-UGG, trnQ-UUG, trnR-ACG, trnR-UCU, trnS-GCU, trnS-GGA, trnS-UGA, trnT-GGU, trnT-UGU, trnV-GAC, trnV-UAC*\*, *trnW-CCA, trnY-GUA* |
| Transcription and splicing | *rpoA, rpoB, rpoC1*\*, *rpoC2, matK* |
| Translation, ribosomal proteins | |
| Initiation factor | *InfA* [!] |
| Small subunit | *rps2, rps3, rps4, rps7, rps8, rps11, rps12*\*\*,T, *rps14, rps15, rps16*\*, *rps18, rps19* |
| Large subunit | *rpl2*\*, *rpl14, rpl16*\*, *rpl20, rpl22*#, *rpl23, rpl32, rpl33, rpl36* |
| Photosynthesis | |
| ATP synthase | *atpA, atpB, atpE, atpF*\*, *atpH, atpI* |
| Photosystem I | *psaA, psaB, psaC, psaI, psaJ, ycf3*\*\*, *ycf4* |
| Photosystem II | *psbA, psbB, psbC, psbD, psbE, psbF, psbH, psbI, psbJ, psbK, psbL, psbM, psbN, psbT, psbZ* |
| Calvin Cycle | *rbcL* |
| Cytochrome complex | *petA, petB*\*, *petD*\*, *petG, petL, petN* |
| NADH dehydrogenase | *ndhA*\*, *ndhB*\*, *ndhC, ndhD, ndhE, ndhF, ndhG, ndhH, ndhI, ndhJ, ndhK* |
| Other | *clpP1*\*\*, *accD, cemA, ccsA, ycf1, ycf2* |

\* indicates genes containing one intron.

\*\* indicates genes containing two introns.

T indicates trans-splicing.

# indicates absence in Fabaceae.

[!] indicates absence in the rosids clade.

We obtained this table from Mader et al. [39] and we added information about the *infA* gene.

## Identification of highly-variable regions for species discrimination

We performed multiple alignments among species belonging to the same family to evaluate the pairwise similarity at both family and genus level (S4 Table). For Fabaceae, we made two separate alignments because of the 50-kb inversion occurring in most Papilionoideae species. Next, we selected those species that exhibit genetic variation of both up to 5 SNPs in the *matK* coding region and 3 SNPs in the *rbcL* coding region to search for highly variable regions for species identification (S4–S6 Tables).

**Fabaceae.** In the genus *Afzelia*, the conventional plant DNA barcodes (*matK*, *rbcL*, *trnL*, *trnL-trnF*, *trnH-psbA*) do not exhibit sufficient polymorphic variations for identification of all species (S4 Table). In contrast, the *matK* gene allows the discrimination at species level within the genera *Erythrophleum*, and *Gilbertiodendron*, and between *Prioria balsamifera* and *Prioria oxyphylla* (S4 Table).

The nucleotide variability of the intergenic spacer *trnH-psbA* is often determined by the presence of sequence inversions in proximity of the *psbA* gene. This variation occurs among species of the subfamily Detarioideae, in *Guibourtia demeusei* and *Guibourtia pellegriniana* (15 bp), in the subfamily Papilionoideae, within the genera *Bobgunnia* (13 bp), *Millettia* (11 bp), *Pterocarpus* (13 bp), and in the subfamily Caesalpinioideae, in *Cylicodiscus gabunensis* (30 bp). We also observed sequence inversions of 23 bp in the intergenic spacer *ccsA-ndhD* in the genera *Afzelia* and *Gilbertiodendron*, and between *Pericopsis* species (6 bp).

Among the protein-coding genes, the CDS of the *ycf1* gene and the *ndhF* gene, which are both located in the SSC region, present enough polymorphic variation for the discrimination of all Fabaceae species in this study. Moreover, the non-coding regions of the *rps16* gene and the *ndhA* gene exhibit unique SNPs that can be used for the identification of all Fabaceae species in this work. The nucleotide variability of the AT-rich intergenic spacers such as the *rps3-rps19*, which is located in the border between the LSC and the IR1 region, is mainly determined by frequent A to T transversion, and it will not be described further.

In the genus *Afzelia*, the intergenic spacers such as the *trnS-trnG* and *trnT-psbD*, as well as three regions of the *ycf1* coding region, exhibit a high nucleotide diversity (Pi) (Fig 2A and S5 and S6 Tables and S2 File). The genetic region including the *psbM* CDS and the intergenic spacer *psbM-trnD*, as well as the region including the first 300 bp of the *rpl16* intron, the intergenic spacer *rpl16-rps3* and the last 150 bp of the *rps3* CDS, exhibit a high number of SNPs.

The analysis of the nucleotide diversity (Pi) among *Guibourtia coleosperma*, *G. pellegriniana* and *G. tessmannii* shows that the *ycf1* gene is the most SNP-rich among coding regions, while the *rps16* gene is the richest one among intron regions (Fig 2B and S5 and S6 Tables and S3 File). The genetic region containing the first 300 bp of the *rpoB* CDS, and the intergenic spacers *rpoB-trnC*, is the most SNP-rich among *Guibourtia* species. Furthermore, also the genetic region including the intergenic spacer *ndhE-ndhG* and the *ndhG* CDS, the region containing the spacer *trnM-atpE* and the *atpE* CDS, as well as the intergenic spacer *petA-psaJ*, exhibit a discrete nucleotide variability.

In this work, we also retrieved the chloroplast genome of species belonging to the subfamily Papilionoideae (Fabaceae), including the two commercial *Millettia* species (*Millettia laurentii* and *Millettia stuhlmannii*), the two Afrormosia wood species (*Pericopsis elata* and *P. angolensis*), the genus *Bobgunnia*, and the commercial timber species of the genus *Pterocarpus*. Except for *Bobgunnia*, the other genera belonging to this sub-family exhibit a 50-kb inversion in the LSC leading to a consequent inversion of gene orientation in this region. The *matK* gene exhibits a high number of polymorphic sites in *Pterocarpus* species, and between the two commercial *Millettia* species (S4 Table).

In the genus *Bobgunnia*, the *ycf1* gene and the genetic region between the *ndhD* and the *ndhE* genes exhibit the highest polymorphic variability (Fig 3A and S5 and S6 Tables and S4

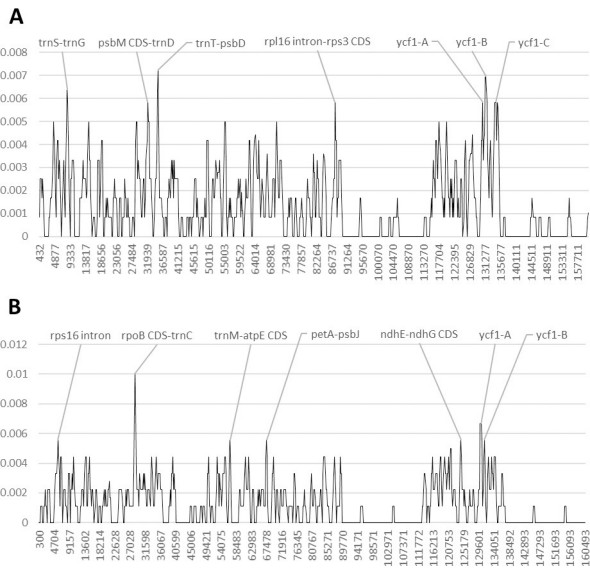

**Fig 2. Nucleotide diversity in Fabaceae, subfamily Detarioideae, upon reversion of sequence inversions.** A) *Afzelia*, B) *Guibourtia*. The y-axis shows the Pi value. The x-axis shows the midpoint value (bp).

File). In addition, the AT-rich intergenic spacer *trnK-rps16* and the region including the 5' extremity of the *ndhG* gene and the AT-rich spacer *ndhG-ndhI* exhibit a high number of polymorphic sites. *Bobgunnia* species have a large AT-rich intergenic spacer located between the IR1 and SSC regions. This region was removed from the alignment because of a high number of A to T transversions.

The analysis of the nucleotide diversity between the two Afrormosia wood species *Pericopsis elata* and *P. angolensis* shows that the genetic region between the *trnT* and *trnD* genes, as well as the intergenic spacer *petN-rpoB* exhibit the highest number of polymorphic sites among

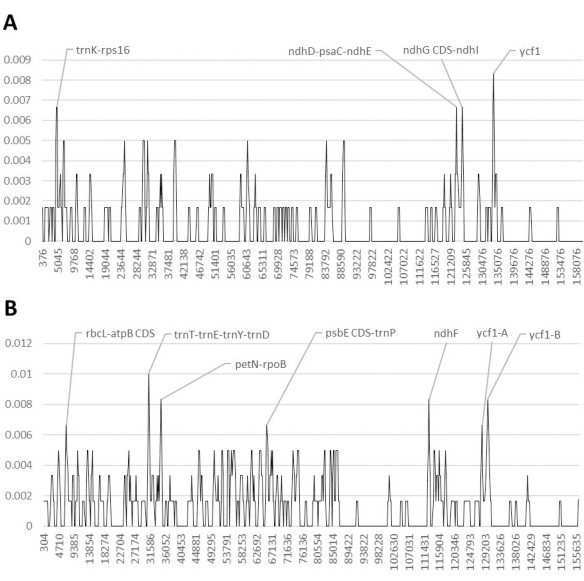

**Fig 3. Nucleotide diversity in Fabaceae, subfamily Papilionoideae, upon reversion of sequence inversions.** A) *Bobgunnia*, B) *Pericopsis*. The y-axis shows the Pi value. The x-axis shows the midpoint value (bp).

non-coding regions (Fig 3B and S5 and S6 Tables and S5 File). Furthermore, the *ndhF* gene and the *ycf1* show the highest nucleotide variability among coding regions. In addition, the genetic region including the intergenic spacer *rcL-atpB* and the *atpB* CDS, as well as the region containing the 5' extremity of the *psbE* CDS and the spacer *psbE-trnP*, also present a moderate genetic variability.

**Meliaceae.**   In the current study, we focused on the discovery of SNPs for the identification of commercial species belonging to the genera *Entandrophragma*, *Khaya*, *Leplaea* (formerly *Guarea*), and *Lovoa*. Genetic regions such as the *ycf1* and the *ndhA* intron show polymorphic variation in all Meliaceae species studied. The conventional plastid markers successfully discriminate between the two *Lovoa* species and among *Entandrophragma* species, as well as between *Leplaea cedrata* and the other *Leplaea* species (S4 Table). We observed a 4-bp sequence inversion in the intergenic spacer *ycf2-trnL* occurring in *Khaya anthotheca* as well as in *Entandrophragma candollei* and *Entandrophragma utile*.

In *Khaya*, we observed no differences in the plastome content between *K. grandifoliola* and *K. senegalensis* (pairwise identity 99.999%). For this reason, we did not include *K. senegalensis* in the analysis of the nucleotide diversity. The intergenic spacers *trnT-psbD* and *psbZ-trnG*, as well as the genetic region between the *ndhF* and the *trnL* genes presents the highest polymorphism rate (Fig 4A and S5 and S6 Tables and S6 File). As for the genes, the coding region of *ndhF*, the *petD* intron and the *atpF* gene are the most variable regions. However, the *ndhF* gene does not exhibit any polymorphic variation between *K. anthotheca* and *K. ivorensis*. In addition, the intergenic spacers *trnK-rps16*, *petN-psbM*, and *ndhC-trnV* also present a discrete polymorphic variation among the three species analyzed.

The conventional plant barcodes exhibit a discrete number of polymorphic sites in *Leplaea cedrata*, but have low or no variability between *L. laurentii* and *L. thompsonii* (S4 Table). The analysis of the nucleotide diversity shows very low genetic differences between *L. laurentii* and *L. thompsonii* (Fig 4B and S5 and S6 Tables and S7 File). The *ccsA* gene exhibits the highest number of polymorphic sites among coding regions. In addition, the genetic region between the *trnQ* and the *psbI* genes, as well as the *rpoC1* gene exhibit a discrete number of polymorphisms.

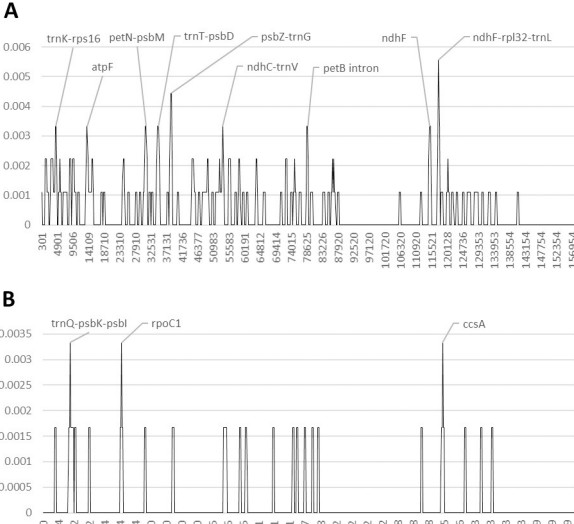

**Fig 4. Nucleotide diversity in Meliaceae upon reversion of sequence inversions.** A) *Khaya*, B) *Leplaea*. The y-axis shows the Pi value. The x-axis shows the midpoint value (bp).

**Sapotaceae.** Within the Sapotaceae, we focused on commercial timber species belonging to the subfamily Sapotoideae, including *Autranella congolensis*, *Baillonella toxisperma*, and *Tieghemella* species, and to the subfamily Chrysophylloideae, including *Pouteria* species (formerly *Aningeria*) and *Gambeya* species (formerly *Chrysophyllum*). The conventional plant DNA barcodes show no polymorphic variation in the genus *Tieghemella* (S4 Table). Interestingly, the *ycf1* coding region in *Tieghemella* exhibits no nucleotide variation as well. In Sapotaceae, we found the same 4-bp inversion in the intergenic spacer *ycf2-trnL*, that we observed in Meliaceae species, in *Tieghemella* species, *Gambeya lacourtiana*, *Pouteria adolfi-friedericii* and *Pouteria aningeri*. In addition, the intergenic spacer *petA-psbJ* exhibits a 16-bp inversion in *Tieghemella africana* and an 18-bp inversion in *Pouteria adolfi-friedericii*.

We performed an analysis of the nucleotide variability among all Sapotoideae species studied, exhibiting high values of pairwise identity, ranging from 99% to 99.9% (S4 Table). Although *Autranella* and *Baillonella* exhibit a good polymorphic variation in the conventional plastid markers, this variation decreases when these two species are compared with *Tieghemella* species (S4 Table). In this group, the *accD*, *ycf1* and *ndhF* genes present the highest nucleotide among coding regions, while the *rpl16* gene exhibits the highest one among intron regions (Fig 5A and S5 and S6 Tables and S8 File). However, only the *ndhF* CDS shows a polymorphic site between *Tieghemella* species. The intergenic spacers *rps16-trnQ* and *ndhF-rpl32* exhibit the highest nucleotide variability. Furthermore, the intergenic spacer *rps16-trnQ*, as well as the genetic region including the intergenic spacer *atpI-rps2* and the *rps2* CDS, are the most SNP-rich regions in the genus *Tieghemella* (Fig 5B and S5 and S6 Tables and S9 File).

We also focused on the analysis of the nucleotide diversity in species belonging to the subfamily Chrysophylloideae. *Gambeya subduna* has sufficient polymorphic variation in the conventional plastid markers when compared to other *Gambeya* species, therefore we did not include it in the analysis of highly variable regions. On the contrary, we found a very low nucleotide variability between *G. gigantea* and *G. laourtiana*. The coding region of the *rpoB* gene and the intergenic spacer *rpl32-trnL* exhibit the highest number of polymorphic sites among *Gambeya* species (Fig 6A and S5 and S6 Tables and S10 File). However, the spacer *rpl32-trnL* does not show any polymorphic variation between *G. gigantea* and *G. lacourtiana*.

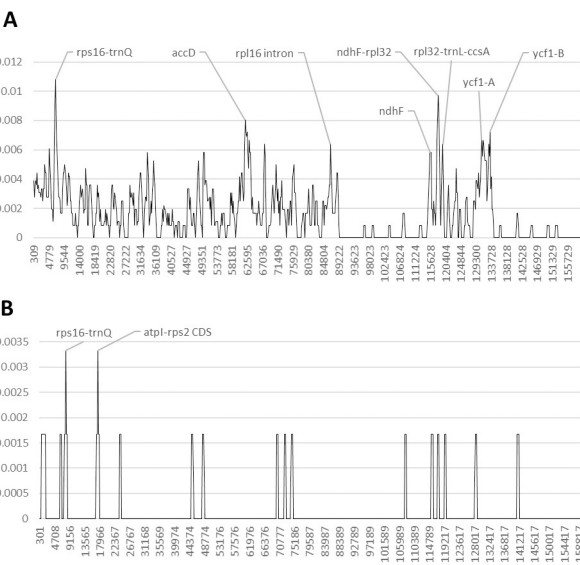

**Fig 5. Nucleotide diversity in Sapotaceae, subfamily Sapotoideae, upon reversion of sequence inversions.** A) Sapotoideae, B) *Tieghemella*. The y-axis shows the Pi value. The x-axis shows the midpoint value (bp).

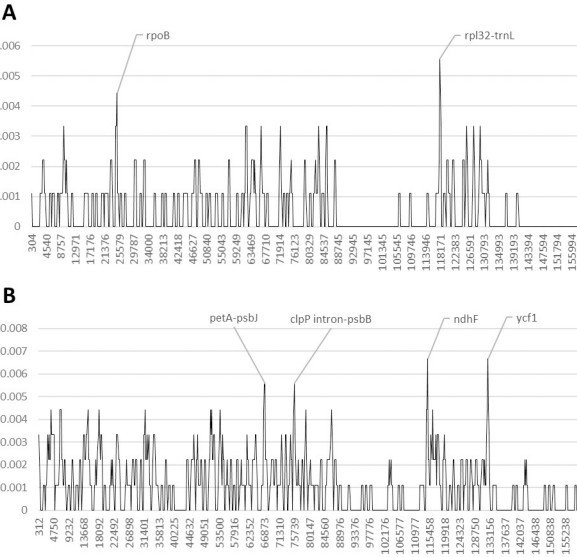

**Fig 6. Nucleotide diversity in Sapotaceae, subfamily Chrysophylloideae, upon reversion of sequence inversions.** A) *Gambeya*, B) *Pouteria*. The y-axis shows the Pi value. The x-axis shows the midpoint value (bp).

In the genus *Pouteria*, the coding regions of the *ndhF* and *ycf1* exhibit the highest number of polymorphic sites (Fig 6B and S5 and S6 Tables and S11 File). In addition, the intergenic spacer *petA-psbJ*, and the genetic region containing the first 250 bp of the *clpP* intron and the spacer *clpP-psbB*, also present a discrete nucleotide variability in *Pouteria* species.

**Other families.** We focused of the search for genetic variations in species of the genera *Milicia* (Moraceae), and *Lophira* (Ochnaceae), and between the two African Ebony species, *Diospyros crassiflora* and *Diospyros mespiliformis* (Ebenaceae). None of the conventional plastid markers shows polymorphic variation between *Lophira* species (S4 Table). Furthermore, both the genera *Milicia* and *Lophira* show low polymorphic variation along the whole plastome sequence (S5 and S6 Tables). On the contrary, the conventional plant DNA barcodes exhibit a high nucleotide variation between the two *Diospyros* species (S4 Table).

In the genus *Milicia*, the coding region of the *rpoA* gene exhibits the highest polymorphic variation (Fig 7A and S5 and S6 Tables and S12 File). In addition, the coding region of the *ycf1* gene and the intergenic spacer *psaI-ycf4* also exhibit a discrete nucleotide variability.

The analysis of the nucleotide diversity in the genus *Lophira* shows that the *ycf1* gene and the region between the *trnE* and the *psbD* genes, present the highest number of polymorphic sites (Fig 7B and S5 and S6 Tables and S13 File).

## Discussion

We reconstructed the chloroplast genomes of 62 commercial timber species from tropical Africa and carried out a comparative analysis to search for highly variable regions that can be used for species identification. Although we observed that the majority of taxa in this study exhibit a strongly conserved gene content, certain genes seem to be lost in some groups. We confirmed that the *infA* gene may be lost in all species from the rosids clade. The loss of the *infA* gene in angiosperms has been previously reported by Millen et al. [43], who found that in some species expressed *infA* copies were transferred to the nucleus. Interestingly, we identified a complete coding sequence of the *infA* gene in the Ebenaceae and Sapotaceae families, belonging to the asterid clade. This provides further evidence to the fact that losses of the *infA* gene in

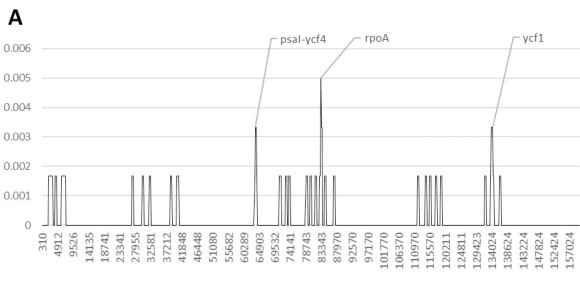

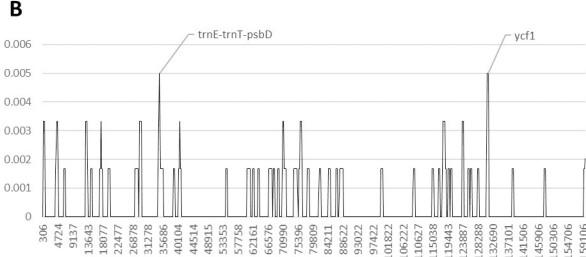

**Fig 7.** Nucleotide diversity in the genera *Milicia* (A) and *Lophira* (B) upon reversion of sequence inversions. The y-axis shows the Pi value. The x-axis shows the midpoint value (bp).

the asterids are parallel 'within-asterid' losses, as probably occurred in common ancestors of Solanaceae and Convolvulaceae [43]. Further investigation is needed to evaluate the effective functionality of the *infA* in Ebenaceae and Sapotaceae, as well as if the *infA* coding region is conserved among species within the Ericales order. Furthermore, we confirmed that the *rpl22* gene was lost in all legumes (Fabaceae). The loss of the *rpl22* may be related to the migration of the gene to the nucleus of a common ancestor of legumes [38].

We also provided evidence that Fabaceae species belonging to the subfamily Papilionoideae exhibit a 50-kb inversion in the plastome, which is a unique feature of this taxonomic group [40]. However, within this subfamily, the papilionoid tribe Swartzieae does not present the 50-kb inversion, as it was still found in *Bobgunnia* species. The lack of the inversion is not surprising because the Swartzieae tribe is a basal clade of the subfamily [40, 41]. Moreover, it elucidates the controversy concerning the placement of the Swartzieae tribe in the Papilionoideae subfamily, previously classified in the Caesalpinioideae subfamily. Besides the 50-kb inversion, we also provided further evidence concerning variations of the gene order which are unique of specific groups in Papilionoideae. We found that species from the genus *Millettia* have a 6.5-kb inversion across the second junction of the 50-kb inversion, between the *trnG-GUG* and the *ycf4* genes. This variation was also found by Kazakoff et al. [42] in *Pongamia pinnata* (formerly *Millettia pinnata*), and might be an intrinsic feature of species belonging to the Millettoid clade. We also found a novel 2-kb inversion in the genus *Pericopsis* between the *psbE* and *psaJ* genes. Previous phylogenetic studies show that the genus *Pericopsis* is closely related to the Ormosieae and Leptolobieae clades, as well as to the genera *Camoensia* and *Clathrotropis* [44, 45]. We observed that the published plastomes of *Ormosia* species [46] do not exhibit the 2-kb inversion between the *psbE* and the *psaJ* genes, supporting the evidence that the 2-kb inversion could be a unique feature of the genus *Pericopsis*. Further molecular investigation on the plastome structure of the closest species to the genus *Pericopsis* is needed to confirm our hypothesis.

The reconstruction of the chloroplast genomes and the multiple alignments were very useful for searching highly variable regions within the whole plastome. This approach allowed us to find genetic differences in species which cannot be distinguished through microscopical

analysis of wood samples. As a result, it led us to select putative genetic markers which may be used for PCR-based identification of species from the international timber market. In this work, we focused on the identification of genetic regions which are rich in *single-nucleotide polymorphisms* (SNPs), similar to the study of Bi et al. [22] on *Fritillaria* species, and that of Song et al. [23] on *Dalbergia* species. We did not take indels into account during the nucleotide diversity analysis as these are mainly consisting of sequence duplications or insertions/deletions in AT-rich or poly-nucleotide regions. In addition, we focused on the detection of sequence inversions in each plastome, yet we re-arranged them before conducting the analysis of highly variable regions, in order to focus exclusively on SNPs.

The multiple alignments revealed the presence of short sequence inversions which often occurred in the species analyzed. For example, these include the intergenic spacers *trnH-psbA* and *ccsA-ndhD* in Fabaceae, the intergenic spacer *ycf2-trnL* in Meliaceae and Sapotaceae, and the *petA-psaJ* in Sapotaceae. We observed that sequence inversions are not intrinsic features at genus or family level, but that they randomly occur within the angiosperms. For this reason, we do not consider sequence inversions as reliable markers for species identification and as such not suitable for DNA barcoding applications. Consequently, sequence inversions should not be used for phylogenetic studies and their application for species identification would present limitations as these do not result in a change of the melting temperature of the target sequence. As a consequence, they cannot be used for species discrimination using techniques such as *Barcode DNA–High-Resolution Melting* (Bar-HRM). On the contrary, the use of SNPs in DNA barcoding applications is advantageous as, when consisting of substitution between adenine/thymine (A/T) and cytosine/guanine (C/G), may be useful for species identification using Bar-HRM.

The traditional plant barcodes such as *matK* and *rbcL* exhibit *single-nucleotide polymorphisms* for the discrimination of a large group of species, while the variability in the intergenic spacer *trnH-psbA* and in the *trnL* intron is mostly determined by gaps in the multiple alignments. Furthermore, the *matK* gene exhibits a high number of polymorphic sites in genera such as *Erythrophleum* and *Pterocarpus* (Fabaceae), *Entandrophragma* and *Lovoa* (Meliaceae), and between the two African Ebony species (*Diospyros*). In contrast, the conventional plastid markers present low or no nucleotide variation in some groups of species in this study. In support of previous studies [15], we confirm that these markers cannot be used as universal markers for the identification of African timber species. For example, none of the standard plastid markers allow the discrimination of all *Afzelia* species and of the two *Tieghemella* and *Lophira* species. As a result, we identified regions exhibiting a higher nucleotide variability than what observed in the conventional plastid markers, which may be used for the development of target sequences for forensic analysis of the wildlife trade. For example, among protein-coding regions, the *ycf1* and the *ndhF* exhibit a high genetic variability among several species in this study. The *ycf1* region was also found to be rich in polymorphic sites in the genera *Fritillaria* [22] and *Dalbergia* [23]. Therefore, the discriminatory power of the *ycf1* and *ndhF* genes, as well as their application in DNA barcoding, should be investigated in a broader group within the angiosperms. On the contrary, we do not recommend to utilize microsatellite regions in families exhibiting a low GC content such as Fabaceae and Moraceae. These microsatellites regions are included in the conventional plant barcode *trnH-psbA*, as well as intergenic spacers such as the *trnK-rps16*, *atpF-atpH-atpI*, *psbZ-trnG*, *rps4-trnL-trnT*, *rbcL-accD*, *rps3-rps19*, *ndhF-rpl32-trnL*, *ndhG-ndhI* and *rps15-ycf1*. Since they present a high AT content and a high number of insertions/deletions, it impacts proper sequencing analysis as well as obscures the identification of effective polymorphic sites.

We noticed that the distribution of polymorphic sites drastically changes among different genera, also belonging to the same family or subfamily. Therefore, we promote the use of the

genome skimming approach for the analysis of large numbers of genetic regions, as well as for the identification of highly variable regions and novel species-specific genetic markers for DNA barcoding in a broader range of species in the whole plant kingdom. Furthermore, the chloroplast genome of a high number of individuals of the same species, with different geographic origins, should be sequenced for a deeper analysis of the interspecific and intraspecific variation, in order to confirm the species-specificity of the selected SNPs. This would lead to the creation of a solid genetic database as a reference for forensic analysis of species from the international timber marker. Then, the whole chloroplast genome would also increase the possibility to develop specific primers to maximize the PCR-based amplification success of the target sequences. These approaches are particularly important for the genetic characterization of those genera exhibiting very low polymorphic variations such as *Khaya* and *Leplaea* (Meliaceae), *Tieghemella* (Sapotaceae), *Milicia* (Moraceae), and *Lophira* (Ochnaceae).

Interestingly, we did not observe any difference between the plastomes of *Khaya grandifoliola* and *Khaya senegalensis*. We hypothesize that the two species may have been subjected to hybridization or incomplete lineage sorting. For this reason, a detailed study of genetic variation among multiple individuals of each species is essential to evaluate the occurrence of these biological processes, which may constitute an issue for the application of plastid markers for species identification. For this, the additional analysis of the nuclear ribosomal internal transcribed spacers *ITS1* and *ITS2* might be useful as a complement to the genomic analysis of chloroplast genomes.

## Conclusions

In summary, we showed that genome skimming is a fast and cost-effective method for the retrieval of sequencing reads and the *de-novo* assembly of chloroplast genomes of a group of land plants using bioinformatics tools. This method also allowed a deeper knowledge of the chloroplast structure, leading to the discovery of differences in the gene order and content in different taxa. The genetic comparison of the plastomes at the family and genus level confirmed that the conventional plastid markers, such as the coding regions of the *matK* and *rbcL* genes, do not always discriminate among land plant species. In addition, we discovered novel SNP-rich genetic regions for the identification of closed-related African timber species of high interest for the international timber market. These regions might be candidate genetic markers for the track and identification of illegal logging activities in tropical Africa, in the framework of CITES, FLEGT, and EU Timber Regulation.

## Supporting information

**S1 Table. Species list, vouchers and citations.**
(XLSX)

**S2 Table. Reference sequences for genome annotation.**
(DOCX)

**S3 Table. Plastome length and GC content.**
(XLSX)

**S4 Table. Pairwise similarity and nucleotide variation in conventional plastid markers.**
(XLSX)

**S5 Table. DNASP6 tables.**
(XLSX)

**S6 Table. List of highly-variable regions.**
(XLSX)

**S7 Table. GenBank accession numbers.**
(XLSX)

**S1 File. List of plastome sequences.**
(FASTA)

**S2 File. Alignment of *Afzelia* species upon reversion of sequence inversions.**
(FASTA)

**S3 File. Alignment of *Guibourtia* species upon reversion of sequence inversions.**
(FASTA)

**S4 File. Alignment of *Bobgunnia* species upon reversion of sequence inversions.**
(FASTA)

**S5 File. Alignment of *Pericopsis* species upon reversion of sequence inversions.**
(FASTA)

**S6 File. Alignment of *Khaya* species upon reversion of sequence inversions.**
(FASTA)

**S7 File. Alignment of *Leplaea* species upon reversion of sequence inversions.**
(FASTA)

**S8 File. Alignment of Sapotoideae species upon reversion of sequence inversions.**
(FASTA)

**S9 File. Alignment of *Tieghemella* species upon reversion of sequence inversions.**
(FASTA)

**S10 File. Alignment of *Gambeya* species upon reversion of sequence inversions.**
(FASTA)

**S11 File. Alignment of *Pouteria* species upon reversion of sequence inversions.**
(FASTA)

**S12 File. Alignment of *Milicia* species upon reversion of sequence inversions.**
(FASTA)

**S13 File. Alignment of *Lophira* species upon reversion of sequence inversions.**
(FASTA)

## Acknowledgments

We thank Sofie D'hondt (Ghent University, Belgium) for supporting us during DNA library preparation. Dr. Olivier Lachenaud (Meise Botanic Garden, Belgium) for the support in the identification of herbarium specimens. Wim Baert and Lynn Delgat (Meise Botanic Garden, Belgium) for supporting us in all experimental steps. All members of the research group of Olivier J. Hardy (*Université Libre de Bruxelles*, Belgium) for the support provided in sampling material and the useful advice. We are also grateful to the staff of various botanical collections for the provision of useful herbarium plant materials: Meise Botanic Garden, Belgium; Herbarium of the *Université Libre de Bruxelles*, Belgium; National Herbarium of The Netherlands

(Naturalis), The Netherlands; Royal Botanic Garden Edinburgh, UK; Royal Botanic Gardens, Kew, UK; Missouri Botanical Gardens, US.

## Author Contributions

**Conceptualization:** Maurizio Mascarello, Mario Amalfi, Olivier J. Hardy, Hans Beeckman, Steven B. Janssens.

**Data curation:** Maurizio Mascarello.

**Formal analysis:** Maurizio Mascarello.

**Funding acquisition:** Steven B. Janssens.

**Investigation:** Maurizio Mascarello, Pieter Asselman.

**Methodology:** Maurizio Mascarello, Pieter Asselman.

**Project administration:** Steven B. Janssens.

**Software:** Maurizio Mascarello.

**Supervision:** Erik Smets, Steven B. Janssens.

**Validation:** Maurizio Mascarello.

**Visualization:** Maurizio Mascarello.

**Writing – original draft:** Maurizio Mascarello.

**Writing – review & editing:** Maurizio Mascarello, Mario Amalfi, Pieter Asselman, Erik Smets, Olivier J. Hardy, Hans Beeckman, Steven B. Janssens.

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
