## [Decision Letter · Decision Letter 0]

4 Mar 2021

PONE-D-21-04340

Genome skimming reveals novel plastid markers for the molecular identification of illegally logged African timber species

PLOS ONE

Dear Dr. Mascarello,

Thank you for submitting your manuscript to PLOS ONE. After careful consideration, we feel that it has merit but does not fully meet PLOS ONE’s publication criteria as it currently stands. Therefore, we invite you to submit a revised version of the manuscript that addresses the points raised during the review process.

We look forward to receiving your revised manuscript.

Kind regards,

Tzen-Yuh Chiang

Academic Editor

PLOS ONE

Journal Requirements:

Reviewers' comments:

Reviewer's Responses to Questions

**Comments to the Author**

1. Is the manuscript technically sound, and do the data support the conclusions?

Reviewer #1: Partly

2. Has the statistical analysis been performed appropriately and rigorously? 

Reviewer #1: N/A

3. Have the authors made all data underlying the findings in their manuscript fully available?

Reviewer #1: Yes

4. Is the manuscript presented in an intelligible fashion and written in standard English?

Reviewer #1: Yes

5. Review Comments to the Author

Reviewer #1: This manuscript " Genome skimming reveals novel plastid markers for the molecular identification of illegally logged African timber species" by Prof. Maurizio Mascarello et al., use high-throughput sequencing to obtain the complete de-novo chloroplast genome of 62 commercial African timber species. Then, authors performed a comparative genomic analysis to mine new candidate molecular markers for the discrimination of closely-related tree species. The article is interesting and will be useful in wildlife forensic species identification. Unfortunately, analysis and interpretation of the data in my opinion is not well done. Authors overemphasized the polymorphisms for these timber species, and only compare the variations between related species. For the application of forensic species identification, author should use these cp genome data to develop markers for species identification. I thought that authors should consider to develop the InDel markers, it will be a fast procedure for species identification. After developin, the PCR amplification should be performed for validating the InDel markers in related species.

L109 chloroplast genes revised to chloroplast genome.

L113 in the angiosperms

L186 check the value “160774 bp ± 602”?

L195 Table 1 37.4±0.0, 36±0.0, 36.8±0.0

L274 Millettia, be italic

L455-456 How to know paraphyletic with the genus Hymenaea? No phylogenetic tree

L491 No difference between Khaya grandifoliola and Khaya senegalensis. No indel existed between two species?

6. PLOS authors have the option to publish the peer review history of their article (what does this mean?). If published, this will include your full peer review and any attached files.

Reviewer #1: No

---

## [Author Response · Author response to Decision Letter 0]

15 Apr 2021

Dear Prof. Tzen-Yuh Chiang,

Thanks a lot for the feedback given. After careful evaluation and discussion with co-authors, I provided you with the answer to reviewer1 comments in the PDF file "Response to Reviewers", which I have attached in the section "Attach file". I am looking forward to hearing your opinion. Thanks for your time and consideration.

Sincerely,

Maurizio Mascarello

---

## [Decision Letter · Decision Letter 1]

30 Apr 2021

Genome skimming reveals novel plastid markers for the molecular identification of illegally logged African timber species

PONE-D-21-04340R1

Dear Dr. Mascarello,

We’re pleased to inform you that your manuscript has been judged scientifically suitable for publication and will be formally accepted for publication once it meets all outstanding technical requirements.

Kind regards,

Tzen-Yuh Chiang

Academic Editor

PLOS ONE

Additional Editor Comments (optional):

Reviewers' comments:

Reviewer's Responses to Questions

**Comments to the Author**

1. If the authors have adequately addressed your comments raised in a previous round of review and you feel that this manuscript is now acceptable for publication, you may indicate that here to bypass the “Comments to the Author” section, enter your conflict of interest statement in the “Confidential to Editor” section, and submit your "Accept" recommendation.

Reviewer #1: (No Response)

2. Is the manuscript technically sound, and do the data support the conclusions?

Reviewer #1: (No Response)

3. Has the statistical analysis been performed appropriately and rigorously? 

Reviewer #1: (No Response)

4. Have the authors made all data underlying the findings in their manuscript fully available?

Reviewer #1: (No Response)

5. Is the manuscript presented in an intelligible fashion and written in standard English?

Reviewer #1: (No Response)

6. Review Comments to the Author

Reviewer #1: According to the author’s responses, I agreed the point that that SNPs in these highly polymorphic regions provided genetic information for future species identification. Because of low number of gaps (no longer than 1 bp), no relevant InDels were found.

7. PLOS authors have the option to publish the peer review history of their article (what does this mean?). If published, this will include your full peer review and any attached files.

Reviewer #1: No

---

## [Editor Report · Acceptance letter]

2 Jun 2021

PONE-D-21-04340R1 

Genome skimming reveals novel plastid markers for the molecular identification of illegally logged African timber species 

Dear Dr. Mascarello:

I'm pleased to inform you that your manuscript has been deemed suitable for publication in PLOS ONE. Congratulations! Your manuscript is now with our production department. 

Kind regards, 

on behalf of

Dr. Tzen-Yuh Chiang 

Academic Editor

PLOS ONE